# Prior expectations evoke stimulus-specific activity in the deep layers of the primary visual cortex

**Fraser Aitken**[1], **Georgios Menelaou**[1], **Oliver Warrington**[1], **Renée S. Koolschijn**[2], **Nadège Corbin**[1], **Martina F. Callaghan**[1], **Peter Kok**[1]*

1 Wellcome Centre for Human Neuroimaging, UCL Queen Square Institute of Neurology, University College London, London, United Kingdom, 2 Wellcome Centre for Integrative Neuroimaging, University of Oxford, FMRIB, John Radcliffe Hospital, Oxford, United Kingdom

* p.kok@ucl.ac.uk

**Data Availability Statement:** All layer-specific time course files and corresponding analysis code are

## Abstract

The way we perceive the world is strongly influenced by our expectations. In line with this, much recent research has revealed that prior expectations strongly modulate sensory processing. However, the neural circuitry through which the brain integrates external sensory inputs with internal expectation signals remains unknown. In order to understand the computational architecture of the cortex, we need to investigate the way these signals flow through the cortical layers. This is crucial because the different cortical layers have distinct intra- and interregional connectivity patterns, and therefore determining which layers are involved in a cortical computation can inform us on the sources and targets of these signals. Here, we used ultra-high field (7T) functional magnetic resonance imaging (fMRI) to reveal that prior expectations evoke stimulus-specific activity selectively in the deep layers of the primary visual cortex (V1). These findings are in line with predictive processing theories proposing that neurons in the deep cortical layers represent perceptual hypotheses and thereby shed light on the computational architecture of cortex.

## Introduction

Over the last decade, research using techniques as diverse as noninvasive functional magnetic resonance imaging (fMRI) [1–4] and electro- and magnetoencephalography [5–8] in humans, as well as invasive animal electrophysiology [9–13], has revealed that prior expectations strongly modulate sensory processing [14]. However, it is as yet unclear what the neural mechanisms underlying these modulations are. To properly understand these mechanisms, we need to go beyond studying cortical regions as a whole and study the laminar circuits involved in these computations [15–17]. The reason for this is that the different cortical layers have different interregional connectivity patterns, with bottom-up signals predominantly flowing from superficial layers 2/3 to the granular layer 4 of downstream regions and feedback arising from the deep layers 5/6 and targeting agranular layers 1 and 5/6 of upstream regions [18–20]. Therefore, determining which layers are involved in a cortical computation can inform us on the likely sources and targets of these signals.

available on the OSF platform (https://osf.io/k54p3).

**Funding:** This work was supported by a Sir Henry Dale Fellowship jointly funded by the Wellcome Trust and the Royal Society (218535/Z/19/Z) to P. K. R.S.K. is supported by an EPSRC/MRC-funded studentship (EP/L016052/1). The Wellcome Centre for Human Neuroimaging is supported by core funding from the Wellcome Trust (203147/Z/16/Z). The Wellcome Centre for Integrative Neuroimaging is supported by core funding from the Wellcome Trust (203139/Z/16/Z). The funders had no role in study design, data collection and analysis, decision to publish, or preparation of the manuscript.

**Competing interests:** The authors have declared that no competing interests exist.

**Abbreviations:** BBR, boundary-based registration; BOLD, blood oxygen level–dependent; CSF, cerebrospinal fluid; EPI, echo planar imaging; fMRI, functional magnetic resonance imaging; GLM, general linear model; GM, grey matter; GRAPPA, GeneRalized Autocalibrating Partial Parallel Acquisition; ITI, intertrial interval; MPRAGE, Magnetization Prepared Rapid Acquisition Gradient Echo; OLS, ordinary least squares; RBR, recursive boundary registration; ROI, region of interest; SDF, signed distance function; SEM, standard error of the mean; SOA, stimulus onset asynchrony; tSNR, temporal signal-to-noise ratio; V1, primary visual cortex; WM, white matter.

For instance, this known physiology has led to predictive coding theories [21–24] proposing that neurons in the deep cortical layers, which send feedback to upstream regions, represent our current hypotheses of the causes of our sensory inputs. Neurons in the superficial layers, on the other hand, are proposed to inform downstream regions of the mismatch between these hypotheses and current sensory inputs. While these theories are intriguing, and have garnered much excitement in the field [25,26], there has been no direct empirical support of these different proposed roles of the cortical layers.

One recent proposal is that expectations evoke stimulus-specific activity patterns in early sensory cortex [27–30], which, in turn, modulate processing of subsequent sensory inputs [31]. The goal of the current study was to determine which layers of the primary visual cortex (V1) contain such expectation signals and how this differs from the activity evoked by a stimulus presented to the eyes. We hypothesised that merely expecting a stimulus might activate a pattern of activity reflecting the expected stimulus in the deep layers of the visual cortex, which have been proposed to contain perceptual hypotheses [22,24,32,33]. Alternatively, expectations may serve to increase the synaptic gain on expected sensory signals in superficial layers, akin to mechanisms of feature-based attention [34,35].

## Results

We induced expectations by presenting a cue (orange versus cyan circle inside a fixation bull's eye) that predicted the orientation (45˚ versus 135˚) of a subsequently presented grating stimulus (Fig 1A and 1B). This first grating was followed by a second grating which differed slightly in orientation (mean = 3.3˚) and contrast (mean = 6.9%), determined by an adaptive staircase (see Materials and methods). In separate runs of the experiment, human participants ($N$ = 18) performed 2 tasks, judging either the orientation or contrast change between the 2 gratings. Crucially, on 25% of trials, the gratings were omitted (Fig 1C), meaning the screen stayed empty except for the fixation bull's eye, and participants were not required to perform any task. Therefore, on these trials, participants had a highly specific expectation of a certain stimulus appearing, but there was no corresponding input to the eyes.

We noninvasively examined the laminar profile of the activity evoked by these orientation expectations in anatomically defined human V1, using ultra-high field (7T) fMRI with high spatial resolution (0.8-mm isotropic). Layer-specific fMRI is a novel technique that has only recently become feasible due to the submillimetre resolution required. It has been successfully used to study many cognitive processes, such as attention [34–36], working memory [37,38], spatial context [39], perceptual illusions [32,40,41], somatosensation [17], and even language [42]. Encouragingly, results have generally been in good alignment with those obtained from invasive animal studies [43–45].

To examine orientation-specific blood oxygen level–dependent (BOLD) activity, we divided V1 voxels into 2 (45˚-preferring and 135˚-preferring) subpopulations depending on their orientation preference during an independent functional localiser. Layer-specific time courses during the main experiment runs were extracted for both voxel subpopulations. Specifically, we defined 3 equivolume grey matter (GM) layers (superficial, middle, and deep; Fig 2) and determined the proportion of each voxel's volume in these layers (as well as in white matter (WM) and cerebrospinal fluid (CSF)). These layer "weights" were subsequently used in a spatial regression analysis to estimate layer-specific time courses of the BOLD signal in the 2 subpopulations of V1 voxels [32,34,38,46]. This regression analysis served to unmix the signals originating from the different layers, which were potentially mixed within individual voxels. Finally, for both voxel subpopulations, we subtracted the activity of the 3 layers evoked by expecting/seeing the non-preferred orientation from the activity evoked by expecting/seeing

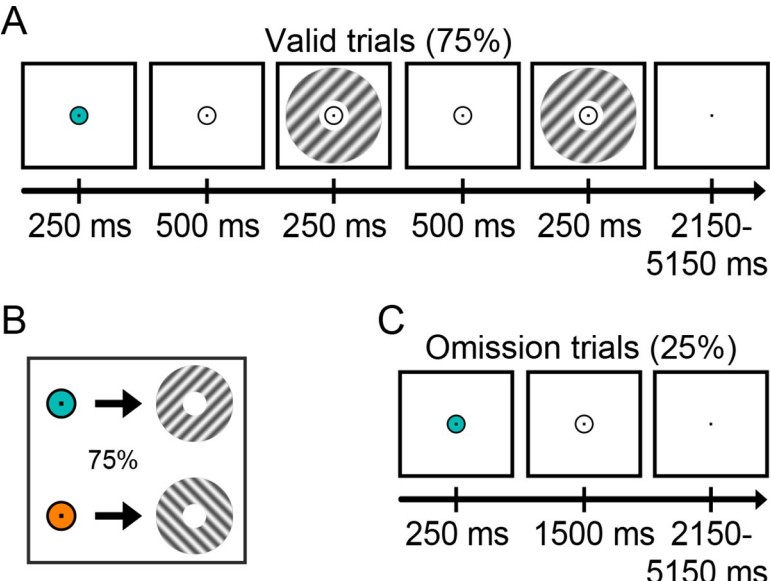

**Fig 1. Experimental paradigm. (A)** Each trial started with a coloured dot that predicted the orientation of the subsequent grating stimulus (45˚ or 135˚). On 75% of trials, a set of gratings was then presented, the first of which had the expected orientation and the second differed slightly in orientation and contrast. In separate experimental runs, participants discriminated either the orientation or the contrast difference between the gratings. **(B)** The colour of the fixation circle (cyan or orange) predicted the orientation of the subsequent grating stimulus (45˚ or 135˚) with 75% validity. **(C)** In the remaining 25% of trials, the gratings were omitted. On these trials, there was an expectation of a particular visual stimulus but no actual visual input. Participants had no task in these trials, except for holding central fixation.

the preferred orientation (see S1 Fig for data before subtraction). After this subtraction, responses were averaged over the 2 subpopulations. This procedure resulted in orientation-specific, layer-specific BOLD signals [34,38] for both the trials on which gratings were presented and those on which the gratings were expected but omitted.

## Prior expectations selectively activate deep layers of V1

The laminar profile of V1 activity evoked by purely top-down expectations, in the absence of bottom-up input, was strikingly different from that evoked by actually presented stimuli (Fig 3A; interaction between presented versus omitted and cortical layer: $F_{2,34}$ = 5.4, $p$ = 0.0093). Specifically, merely expecting a grating with a specific orientation evoked a BOLD signal reflecting that orientation in the deep ($t_{17}$ = 3.5, $p$ = 0.0029), but not the middle ($t_{17}$ = 0.8, $p$ = 0.45) or superficial ($t_{17}$ = 0.7, $p$ = 0.48) layers of V1. In fact, the expectation-evoked orientation-specific BOLD signal was significantly stronger in the deep layers than in the middle ($t_{17}$ = 2.8, $p$ = 0.012) and superficial ($t_{17}$ = 2.5, $p$ = 0.025) layers. When a grating was actually presented to the eyes, this evoked orientation-specific activity in all layers of V1 (deep: $t_{17}$ = 4.5, $p$ = 0.00035; middle: $t_{17}$ = 3.6, $p$ = 0.0022; superficial: $t_{17}$ = 4.2, $p$ = 0.00063), as would be expected.

Interestingly, the strength of the expectation signals was dependent on the task the participants were performing in that experimental run (Fig 3B; interaction between stimulus presented versus omitted, task, and cortical layer: $F_{2,34}$ = 3.3, $p$ = 0.048). This effect was driven by the fact that orientation-specific BOLD signals evoked by expected-but-omitted gratings had a different laminar profile in the 2 tasks (interaction between task and cortical layer, omission trials only: $F_{2,34}$ = 3.6, $p$ = 0.039; Fig 3B, top panel). That is, orientation expectations evoked

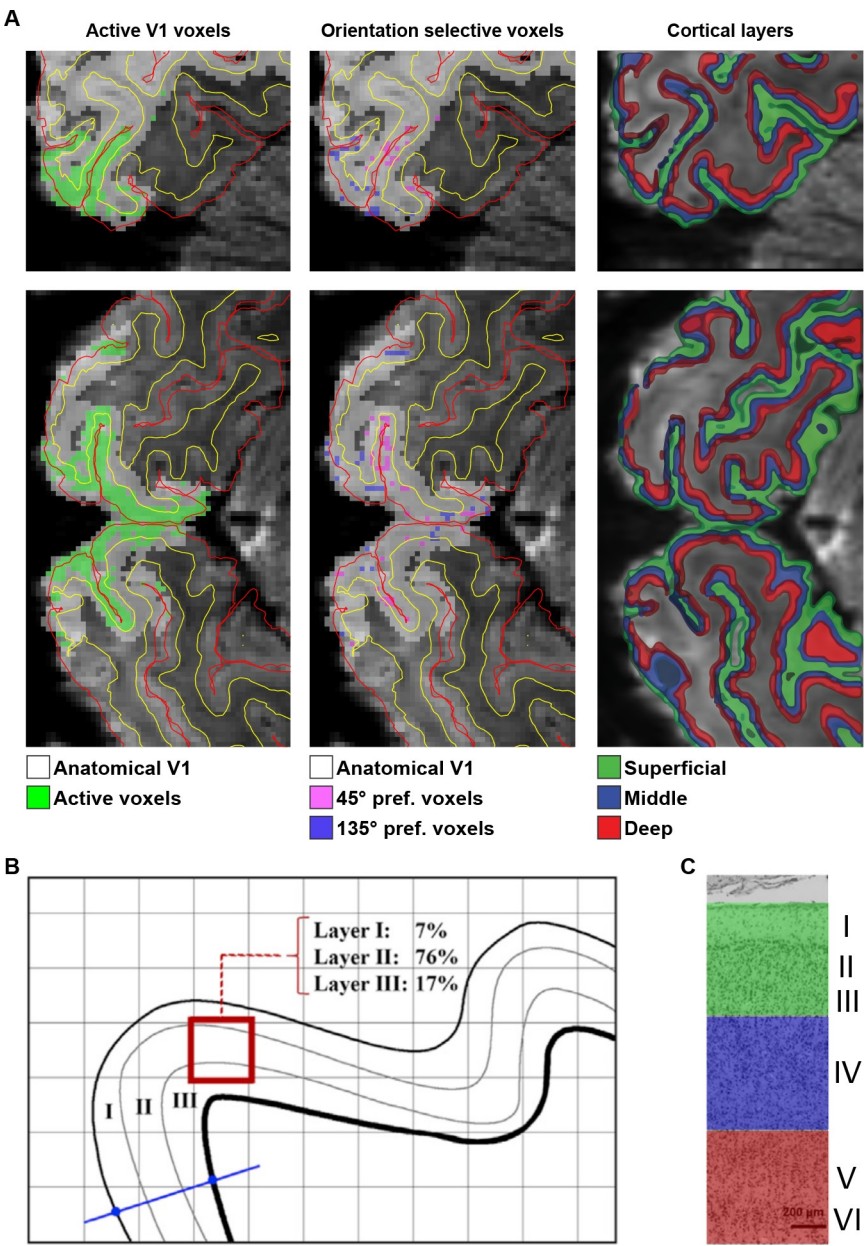

**Fig 2. Analysis approach.** (A) Illustration of ROI selection on sagittal slice of the mean functional scan of 1 participant. Overlaid red and yellow lines indicate pial and WM boundaries, respectively, as determined by registering anatomical boundaries to the mean functional image using RBR. Within V1 (white), active voxels were selected based on significant activation in the functional grating localiser (green). From these active voxels, we selected the 500 most strongly 45˚-preferring (pink) and 135˚-preferring (blue) voxels, respectively. With all voxels in these 2 ROIs, we determined how their volume was distributed over the superficial, middle, and deep cortical layers. (B) Schematic example of a voxel (red square) and the distribution of its volume over the 3 GM layers. This layer volume distribution was determined for each voxel and used as the basis of a regression approach in order to obtain layer-specific BOLD time courses (see Materials and methods). (C) Deep, middle, and superficial cortical layers indicated in coloured ribbons. Cytoarchitectural image of V1 adapted from [88]. BOLD, blood oxygen level–dependent; GM, grey matter; RBR, recursive boundary registration; ROI, region of interest; V1, primary visual cortex; WM, white matter.

stimulus-specific activity in the deep ($t_{17}$ = 3.0, $p$ = 0.0086), but not the middle ($t_{17}$ = 1.1, $p$ = 0.28; deep versus middle: $t_{17}$ = 2.6, $p$ = 0.020) and superficial ($t_{17}$ = 0.1, $p$ = 0.92; deep versus superficial: $t_{17}$ = 3.5, $p$ = 0.0026) layers of V1 when participants were preparing to perform

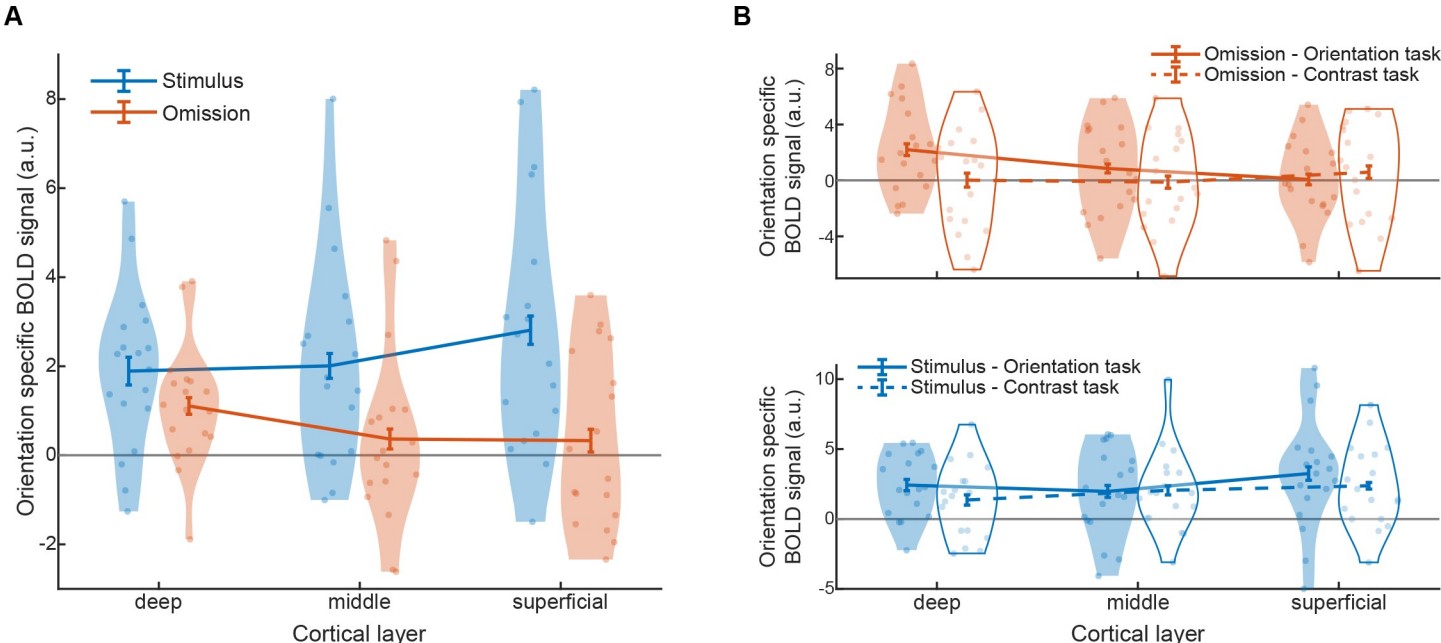

**Fig 3. Layer-specific BOLD response in V1 for presented and expected stimuli. (A)** Orientation-specific BOLD response to presented (blue) and expected-but-omitted (orange) gratings in the different layers of V1, averaged over tasks. **(B)** Orientation-specific BOLD response to expected-but-omitted (orange, top panel) and presented (blue, bottom panel) gratings, separately for the orientation (solid lines, filled shapes) and contrast (dashed lines, open shapes) tasks. Dots represent individual participants, and curved shapes indicate density. Error bars indicate within-subject SEM. Data are available at osf.io/k54p3. BOLD, blood oxygen level–dependent; SEM, standard error of the mean; V1, primary visual cortex.

an orientation discrimination, while this effect was absent when they were preparing to discriminate the contrast of the gratings (deep: $t_{17} < 0.1$, $p = 0.98$, middle: $t_{17} = -0.2$, $p = 0.88$, superficial: $t_{17} = 0.7$, $p = 0.49$). Though it should be noted that the differences in omission-evoked activity between the 2 tasks were not significant in any of the layers (deep: $t_{17} = 1.5$, $p = 0.16$; middle: $t_{17} = 0.8$, $p = 0.44$; superficial: $t_{17} = -0.4$, $p = 0.68$). The laminar profile of stimulus-evoked (rather than omission-evoked) activity did not differ between the 2 tasks (interaction between task and cortical layer, stimulus present trials only: $F_{2,34} = 1.8$, $p = 0.18$; Fig 3B, bottom panel). Given that accuracy and reaction times did not differ between the 2 tasks (accuracy: 79.6% versus 78.9%; $t_{17} = 0.4$, $p = 0.72$; RT: 689 ms versus 676 ms; $t_{17} = 1.2$, $p = 0.23$), the difference in the laminar profiles evoked by expected-but-omitted gratings are unlikely to be due to task difficulty or engagement, but more likely due to the fact that the expected feature, orientation, was task relevant in 1 task but not the other. We are cautious to overinterpret this effect given that some previous studies with similar experimental designs have not shown it [1,29], but this could indicate that expectation signals are stronger when they pertain to a task-relevant or attended feature [47].

## Control analyses

**Balancing the laminar design matrices.** We found that, on average, a larger proportion of the volume of the voxels in our region of interests (ROIs) overlapped with the superficial (28 +/− 3%, mean +/− SD over participants) than the middle (21% +/− 2%) and deep (16% +/− 1%) layers. This is likely the result of the well-known bias towards superficial layers in the BOLD signal as measured with gradient echo sequences due to venous blood draining from the deeper layers towards the surface [48,49]. While such a bias is unlikely to explain our main

results, which in fact reveal a dominance of the deep layers in activity evoked by prior expectations, we nevertheless repeated our analyses after correcting for this imbalance. That is, for each ROI, we retained a subset of the selected voxels such that all layers were represented equally, by iteratively removing voxels from the most overrepresented layer until a 1-way ANOVA on the remaining voxels in a given ROI (with independent variable "layer") no longer revealed a significant ($p > 0.1$) imbalance. This resulted in the exclusion of 80 (+/− 31, mean +/− SD) voxels per ROI on average. This control analysis yielded qualitatively the same effects as our main analysis. That is, the laminar profile of orientation-specific BOLD signal evoked by expectations, in the absence of bottom-up input, was significantly different from that evoked by actually presented gratings ($F_{2,34} = 4.62$, $p = 0.017$). Specifically, expectations evoked orientation-specific BOLD signal in the deep ($t_{17} = 3.32$, $p = 0.0040$), but not in the middle ($t_{17} = 1.29$, $p = 0.21$) and superficial ($t_{17} = 0.10$, $p = 0.92$) layers. As expected, actually presented gratings evoked activity in all layers of V1 (deep: $t_{17} = 4.39$, $p = 0.00040$; middle: $t_{17} = 3.69$, $p = 0.0018$; superficial: $t_{17} = 3.41$, $p = 0.0034$).

**Results based on raw voxel time courses.** For the main analysis, individual voxel time courses were normalised and weighted by how orientation selective they were (see Materials and methods for details) before applying the layer extraction. In a control analysis, we conducted our analyses on the raw voxel time courses, omitting the normalisation and weighting steps. This analysis qualitatively reproduced our main results: The laminar profile of orientation-specific BOLD signal evoked by expected-but-omitted gratings was significantly different from that evoked by actually presented gratings ($F_{2,34} = 8.68$, $p = 0.00090$; S2 Fig). As in our main analysis, expectations evoked orientation-specific BOLD signal in the deep ($t_{17} = 3.30$, $p = 0.0042$), but not in the middle ($t_{17} = 0.21$, $p = 0.84$) and superficial ($t_{17} = 0.22$, $p = 0.82$) layers, while actually presented gratings evoked activity in all layers of V1 (deep: $t_{17} = 4.13$, $p = 0.00070$; middle: $t_{17} = 2.88$, $p = 0.010$; superficial: $t_{17} = 3.71$, $p = 0.0017$).

**Estimating layer-specific time courses through interpolation rather than spatial GLM.** To rule out that our results were driven by the specifics of the laminar regression method, we repeated our main analyses on layer-specific time courses extracted through interpolation rather than a spatial general linear model (GLM; see Materials and methods for details). As for the other control analyses, this yielded qualitatively the same results as our main analysis: The laminar profile of orientation-specific BOLD signal evoked by expected-but-omitted gratings was significantly different from that evoked by actually presented gratings ($F_{2,34} = 9.11$, $p = 0.00068$; S3 Fig). This was driven by the fact that expected-but-omitted gratings evoked orientation-specific BOLD signal in the deep ($t_{17} = 2.74$, $p = 0.014$), but not in the middle ($t_{17} = 1.17$, $p = 0.26$) and superficial ($t_{17} = 1.00$, $p = 0.33$) layers, while actually presented gratings evoked activity in all layers of V1 (deep: $t_{17} = 4.23$, $p = 0.00057$; middle: $t_{17} = 4.13$, $p = 0.00070$; superficial: $t_{17} = 4.27$, $p = 0.00052$).

**Voxel selection.** The main results were based on the 500 V1 voxels most selective for 45˚ and 135˚ oriented gratings, respectively. In order to establish whether the results depended on the exact number of voxels selected, we conducted a control analyses which quantified the orientation-selective BOLD response evoked by presented and expected-but-omitted gratings as a function of the number of selected voxels, ranging from 100 to 1,000 voxels per orientation. Even though the orientation-selective BOLD response overall decreased as more less-selective voxels were added, the main results were qualitatively similar over the range of selected voxels (S4 Fig), establishing that the layer-specific effects of expectation did not depend on the exact number of selected voxels.

## Discussion

In short, we found that prior expectations evoke stimulus-specific activity selectively in the deep layers of V1. Interestingly, this selective activation of the deep layers matches that evoked by illusory Kanizsa figures, which have been suggested to be the result of an automatic structural expectation in the visual system [32]. However, since the expectations in the current experiment were signalled by a conditional cue, V1 activity is more likely to be the result of feedback from higher-order regions outside of the visual cortex, possibly involving the hippocampus [27,50–52]. Speculatively, the fact that these very different types of "expectations" evoke highly similar layer-specific activity in visual cortex may point to a common computational role in V1.

Cortico–cortical feedback connections preferentially target layers 1 and 5/6 in upstream regions [19,20]. While feedback connections to layers 5/6 may explain our current findings, feedback signals in layer 1 may be more difficult to detect directly using fMRI, since layer 1 is very thin and sparsely populated with neurons. Rather, feedback connections to layer 1 likely modulate activity in other layers, by targeting dendritic tufts of layers 2/3 and 5 pyramidal cells extending into layer 1 [19]. These feedback connections, which target apical dendritic tufts far from the soma, are unlikely to drive activity in pyramidal neurons by themselves, but have rather been suggested to have a modulatory function, such as coincidence detection [53]. However, in the omission trials in the current study, there were no bottom-up sensory signals to be modulated. This might explain why we here observe (driving) effects of feedback to deep layers, but no (modulatory) effects in the superficial layers of V1. This is in line with previous findings where feedback in the absence of bottom-up input activated only the deep layers of V1 (as here), but feedback in the presence of bottom-up input led to modulations in both deep and superficial layers [32]. Therefore, we hypothesise that if, instead of omitting the predicted stimuli, one would present unexpected stimuli, this would lead to modulations in both deep and superficial layers. Future studies will be needed to test this hypothesis.

One proposed neural implementation of perceptual inference is predictive coding [22,23], a theory that proposes that each cortical region houses separate subpopulations of neurons coding for perceptual hypotheses (predictions) and mismatches between these hypotheses and bottom-up sensory input (prediction errors) [10,11]. Since feedback mainly arises from the deep layers, prediction units are suggested to reside predominantly in the deep layers, while prediction error units dominate in the middle and superficial layers [21,24]. The current results are in line with this proposed arrangement, since a prediction in the absence of any bottom-up input was found to evoke stimulus-specific signals in only the deep layers of V1. However, it is important to point out that our findings are not exclusive support for predictive coding theories but are in line with other theories that suggest that predictive feedback plays a crucial role in sensory processing as well (e.g., [33,54]). In future work, it will be crucial to test whether or not cortex indeed calculates explicit prediction errors during sensory processing, as proposed by predictive coding theories.

Notably, stimulus-specific V1 activity evoked by maintaining a grating stimulus in working memory has a strikingly different laminar profile, activating both the deep and the superficial (but not the middle) layers [38]. The MRI data acquisition and analysis methods in the current study closely matched those used in the study by Lawrence and colleagues [38], rendering an explanation in terms of methodological differences unlikely. The recruitment of the deep layers by both expectation and working memory could represent the use of internally generated stimulus representations in both processes. In addition, the conscious effort of maintaining a stimulus in working memory for goal-directed behaviour may require feature-based attention, which has been suggested to modulate activity in the superficial layers [34,35]. Therefore,

speculatively, working memory may be a compound process, activating deep and superficial layers for different computational reasons. More broadly, it has been suggested that expectation and attention may be separable neural processes [55–57], with distinct computational roles: activating hypotheses and boosting relevant inputs, respectively [58,59]. Layer-specific fMRI offers the opportunity to directly address these questions in the human brain for the first time [16].

Recent work has revealed that mental imagery can also induce stimulus-specific signals selectively in the deep layers of V1 [60]. This suggests that perhaps the crucial factor determining the involvement of the superficial layers in top-down modulations lies in whether they involve keeping a presented stimulus online (as in most working memory tasks) or generating a stimulus presentation de novo (as in mental imagery tasks and in the omission trials in the current study). Further research will be needed to address this question, as well as determine whether or not mental imagery and expectation involve distinct neural mechanisms [61].

Layer-specific fMRI is a novel technique that is not without its challenges, such as the nonlinear relationship between layer-specific neural activity and the BOLD signal, due to the complex microvasculature of cortex (e.g., draining veins). For this reason, it is crucial to compare laminar profiles between experimental conditions that isolate a specific effect of interest, since a single laminar BOLD profile can be challenging to interpret. Encouragingly, results from studies that have done so have generally been in good alignment with invasive animal electrophysiology [43–45]. For instance, Lawrence and colleagues [38] reported effects of working memory in deep and superficial layers of V1, in line with macaque studies [62], while stimulus contrast most strongly modulated responses in the middle layers [34], as would be expected given that layer 4 is the main bottom-up input layer. Finally, Fracasso and colleagues [43] reported that estimated V1 centre and surround receptive field sizes are smallest in the middle layers and larger in agranular layers, in line with reports of macaque V1 [63]. Additionally, efforts are ongoing to correct for known effects of vasculature using more sophisticated analysis techniques [48,64]. Further research will be needed to corroborate the results from human layer-specific fMRI studies using invasive electrophysiology. Human neuroimaging and invasive neurophysiology are highly complementary approaches that often do not sufficiently interact, and layer-specific fMRI and other high-resolution neuroimaging methods provide an excellent opportunity to start bridging this gap.

It is interesting to note that the presence of the expectation signals depended on the task the participants were performing, suggesting that expectation signals were stronger when they pertained to an attended feature (i.e., during the orientation task) than to an unattended feature (during the contrast task) [47]. (Though note that the deep layer signals were not significantly stronger in the orientation task than in the contrast task, prompting caution in interpreting these effects.) This dependency was not found in recent studies using a very similar experimental design investigating expectation signals using magnetoencephalography [29] and effects of expectation on stimulus processing in V1 using fMRI [1]. Furthermore, several previous studies have reported early sensory signals evoked by task-irrelevant predictions [30,65,66]. One possibility is that these previous studies simply were not able to detect these task modulations because they lacked laminar resolution; note that the task dependence here is expressed as an interaction between task and cortical layer. An alternative possibility is that attention may boost the gain of expectation signals, perhaps even promoting them from activity-silent synaptic plasticity to being reflected in neural firing [67,68], but that this does not change the consequences of these modulations for subsequent stimulus processing [1]. Future layer-specific research orthogonally manipulating attention and expectation validity will be needed to distinguish these possibilities.

How might prediction signals in the deep layers modulate processing of incoming sensory inputs? One potential mechanism for this is through inhibitory connections from the deep layers to the middle and superficial layers [19,69–71], which might cause a reduction in activity throughout the entire cortical column [72] as a result of the excitatory pathways from layer 4 to layers 2/3 and from there to layers 5/6 [18,19]. In the absence of sensory input to layer 4, as is the case in the omission trials here, this modulation would not occur, and top-down feedback signals would be restricted to the deep layers. An alternative mechanism could be through feedback connections terminating on inhibitory neurons in layer 1, which, in turn, inhibit pyramidal neurons in layers 2/3 [21,73,74]. Possibly in line with this, a recent laminar fMRI study of contextual effects in non-stimulated human V1 reported effects in the most superficial layers, potentially reflecting layer 1 [39].

The finding that prior expectations evoke stimulus-selective signals selectively in the deep layers of visual cortex sheds light on the neural circuit by which the brain performs perceptual inference [21,24,33], that is, combines sensory signals with internal expectations to generate a best guess of what is out there in the world. Ultimately, future work building on these findings may be able to reveal how this delicate balance between internal and external signals can go awry, as it does in disorders such as autism [75–77] and psychosis [16,78,79].

## Materials and methods

### Ethics statement

The study was approved by the Oxford University Medical Sciences Interdivisional Research Ethics Committee (R61495/RE001) and was conducted according to the principles of the Declaration of Helsinki. All participants gave written informed consent prior to participation and received monetary compensation.

### Participants

Twenty-three healthy human volunteers with normal or corrected-to-normal vision participated in the 7T fMRI experiment. One participant was excluded because they responded to <50% of trials during the fMRI session. One further participant was excluded because the calcarine sulcus was not in the field of view for the entire fMRI session due to large head movements between runs. Finally, 3 participants were excluded due to our strict head motion criteria of no more than 10 movements larger than 1.0 mm in any direction between successive functional volumes. For the remaining participants, the maximum change in head position in any direction over the course of the fMRI runs was within 4 mm (1.9 +/− 0.8 mm, mean +/− SD over participants) of the mean head position (to which the anatomical boundaries were registered). The final sample consisted of 18 participants (10 female; age 25 ± 4 years; mean ± SD).

### Stimuli

Grayscale luminance-defined sinusoidal Gabor grating stimuli were generated using MATLAB (MathWorks, Natick, Massachusetts, United States of America, RRID:SCR_001622) and the Psychophysics Toolbox [80]. During the behavioural session, the stimuli were presented on a MacBook Pro (Apple, Cupertino, California, USA; 1280 × 800 screen resolution, 60-Hz refresh rate). In the fMRI scanning session, stimuli were projected onto a rear projection screen using an Eiki LC-XL100 projector with custom throw lens (Eiki Industrial, Itami, Hyogo, Japan; 1024 × 768 screen resolution, 60-Hz refresh rate) and viewed via a mirror (view distance 60 cm). Visual prediction cues consisted of a circular region within a white fixation bull's eye

(0.7˚ diameter) turning either cyan or orange for 250 ms. On valid trials (75%), cues were followed by a set of 2 gratings (1.5-cpd spatial frequency, 250-ms duration each, and separated by a 500-ms blank screen), displayed in succession in an annulus (outer diameter: 10˚ of visual angle, inner diameter: 1˚, contrast decreasing linearly to 0 over 0.7˚ at the inner and outer edges), surrounding a fixation bull's eye (0.7˚ diameter). The central fixation bull's eye was presented throughout the trial, as well as during the intertrial interval (ITI; jittered exponentially between 2,150 and 5,150 ms).

## Experimental procedure

Trials consisted of a coloured prediction cue, followed by 2 consecutive grating stimuli on 75% of trials (750-ms stimulus onset asynchrony (SOA) between cue and first grating) (Fig 1A). The coloured cue (cyan or orange) predicted the orientation of the first grating stimulus (45˚ or 135˚) (Fig 1B). On valid trials (75%), 2 consecutive grating stimuli were presented following the coloured cue. The first grating had the orientation predicted by the cue (45˚ or 135˚) and a luminance contrast of 80%. The second grating differed slightly from the first in terms of both orientation and contrast (see below), as well as being in antiphase to the first grating (which had a random spatial phase). On the remaining 25% of trials, no gratings were presented (omission trials; Fig 1C), and the screen remained empty except for the fixation bull's eye. Participants had no task on these trials, except for holding central fixation. The contingencies between the cue colours and grating orientations were flipped halfway through the experiment (i.e., after 2 runs), and the order was counterbalanced over participants.

In separate runs (2 blocks of 64 trials each, approximately 13 minutes), participants performed either an orientation or a contrast discrimination task on the 2 gratings. When performing the orientation task, participants had to judge whether the second grating was rotated clockwise or anticlockwise with respect to the first grating. In the contrast task, a judgement had to be made on whether the second grating had lower or higher contrast than the first one. These tasks were explicitly designed to avoid a direct relationship between the perceptual expectation and the task response. Furthermore, these 2 different tasks were designed to manipulate the task relevance of the grating orientations, to investigate whether the effects of orientation expectations depend on the task relevance of the expected feature. Participants indicated their response (response deadline: 750 ms after offset of the second grating) using an MR-compatible button box. The orientation and contrast differences between the 2 gratings were determined by an adaptive staircase procedure [81], being updated after each trial. This was done to yield comparable task difficulty and performance (approximately 75% correct) for the different tasks. Staircase thresholds obtained during 1 task were used to set the stimulus differences during the other task in order to make the stimuli as similar as possible in both contexts.

All participants completed 4 runs (2 of each task, alternating every run, order was counterbalanced over participants) of the experiment, yielding a total of 512 trials, 256 per task. At the start of each block, the relationship between the cue and the stimulus was shown by presenting the predicted orientation within an appropriately coloured circle. The staircases were kept running throughout the experiment. Prior to entering the scanner, as well as in between runs 2 and 3, when the contingencies between cue and stimuli were flipped, participants performed a short practice run containing 32 trials of both tasks (approximately 4.5 minutes).

Participants underwent a behavioural practice session just prior to entering the scanner to ensure knowledge of the task and how to respond. In the practice session, participants were given written and verbal instructions about the task requirements. During the practice runs, the coloured cues predicted the orientation of the first grating stimulus of the pair with 100% validity (45˚ or 135˚; no omission trials).

After the main experiment, participants performed a functional localiser task inside the scanner. This consisted of flickering gratings (2 Hz), presented at 100% contrast, in blocks of approximately 14.3 seconds (4 TRs). Each block contained gratings with a fixed orientation (45˚ or 135˚). The 2 orientations were presented in a pseudorandom order followed by an approximately 14.3-second blank screen, containing only a fixation bull's eye. Participants were tasked with responding whenever the white fixation dot briefly dimmed to ensure central fixation. All participants were presented with 16 localiser blocks.

## fMRI data acquisition

Functional images were acquired on a Siemens Magnetom 7T MRI system (Siemens Healthcare GmbH, Erlangen, Germany) with a single-channel head coil for localised transmission with a 32-channel head coil insert for reception (Nova Medical, Wilmington, USA) at the Wellcome Centre for Integrative Neuroimaging (University of Oxford) using a $T2^*$-weighted 3D gradient-echo EPI sequence (volume acquisition time of 3,583 ms, TR = 74.65 ms, TE = 29.75 ms, voxel size $0.8 \times 0.8 \times 0.8$ mm, 16˚ flip angle, field of view $192 \times 192 \times 38.4$ mm, in-plane GeneRalized Autocalibrating Partial Parallel Acquisition (GRAPPA) acceleration factor 4, in-plane partial Fourier 6/8, echo spacing 1.27 ms). Anatomical images were acquired using a Magnetization Prepared Rapid Acquisition Gradient Echo (MPRAGE) sequence (TR = 2,200 ms, TE = 2.96 ms, TI = 1,050 ms, voxel size $0.7 \times 0.7 \times 0.7$ mm, 7˚ flip angle, field of view $224 \times 224 \times 179.2$ mm, in-plane GRAPPA acceleration factor 2).

## Preprocessing of fMRI data

The first 4 volumes of each run were discarded. The functional volumes were cropped to cover only the occipital lobe to reduce the influence of severe distortions in the frontal lobe. The cropped functional volumes were spatially realigned within scanner runs, and subsequently between runs, to correct for head movement using SPM12. Temporal signal-to-noise ratio (tSNR, defined as mean signal/SD over time) of in-brain voxels was significantly higher after (12.5 +/− 1.6, mean +/− SD over participants) than before (9.2 +/− 1.6) spatial realignment. FSL FAST [82] was used to correct the bias field and remove intensity gradients in the MPRAGE anatomical image.

## Segmentation and coregistration of cortical surfaces

Freesurfer (http://surfer.nmr.mgh.harvard.edu/) was used to detect boundaries between grey and WM and CSF, respectively, on the basis of the bias-corrected MPRAGE. Manual corrections were made to remove dura incorrectly included in the pial surface when necessary. The GM boundaries were registered to the mean functional volume in 2 steps: (1) a conventional rigid body boundary-based registration (BBR) [83]; and (2) recursive boundary registration (RBR) [84] (S5 Fig). During RBR, BBR was applied recursively to increasingly smaller partitions of the cortical mesh. Here, we applied affine BBR with 7 degrees of freedom: rotation and translation along all 3 dimensions and scaling along the phase-encoding direction. In each iteration, the cortical mesh was split into 2, and the optimal BBR transformations were found and applied to the respective parts. Subsequently, each part was split into 2 again and registered. The specificity increased at each stage and corrected for local mismatches between the structural and the functional volumes that are due to magnetic field inhomogeneity-related distortions. Here, we ran 6 such iterations. The splits were made along the cardinal axes of the volume, such that the number of vertices was equal for both parts. The plane for the second cut is orthogonal to the first, the third orthogonal to the first 2. The median displacement was

taken after running the recursive algorithm 6 times, in which different splitting orders where used, comprised of all 6 permutations of x, y, and z.

## Definition of regions of interest

The V1 surface label, defined by Freesurfer based on the anatomy of the MPRAGE image, was projected into volume space covering the full cortical depth plus a 50% extension into WM and CSF, respectively. The V1 ROI was constrained to only the most active voxels in response to the grating stimuli by applying a temporal GLM to the preprocessed data from the functional localiser run. Blocks of 45˚ and 135˚ gratings were modelled separately as regressors and contrasted together against baseline to identify voxels that exhibited a significant response to the grating stimuli ($t > 2.3$, $p < 0.05$; 6,009 +/− 2,033 voxels, mean +/− SD over participants). To estimate the orientation preference of each voxel, the 2 orientation regressors were contrasted against each other. We selected the 500 voxels with the most positive T values (45˚ preferring) and the 500 voxels with the most negative T values (135˚ preferring) in the contrast and created separate masks for each. Finally, we z-scored the time course data from each voxel and multiplied this time course with the absolute T-value from the orientation contrast (45˚ versus 135˚) in order to weight the results towards the voxels with the most robust orientation preference [34,38]. These ROI definitions were identical to those used in previous studies that successfully resolved orientation-specific BOLD signals with layer specificity [34,38]. We matched our analysis approach to these previous studies to be able to compare our effects of prior expectations on orientation-specific BOLD signals to the effects of working memory [38] and attention [34].

## Definition of the cortical layers

GM was divided into 3 equivolume layers using the level set method (described in detail in [46,85,86]) following the principle that the layers of the cortex maintain their volume ratio throughout the curves of the gyri and sulci [87]. Briefly, the level set function is a signed distance function (SDF), where points on the same surface equal 0 and values on 1 side of the surface are negative and values on the other are positive. The level set function for the GM–CSF and GM–WM boundaries is calculated, and then intermediate surfaces can be defined by moving the surface to intermediate cortical depths. The equivolume model transforms a desired volume fraction into a distance fraction, taking the local curvature of the pial and WM surfaces at each voxel into account [46]. Here, we calculated 2 intermediate surfaces between the WM and pial boundaries, yielding 3 GM layers (deep, middle, and superficial). In human V1, these 3 laminar compartments are expected to correspond roughly to layers I to III, layer IV, and layers V and VI, respectively [88] (Fig 2C). Based on these surfaces, we calculated 4 SDFs, containing for each functional voxel its distance to the boundaries between the 5 cortical compartments (WM, CSF, and the 3 GM layers). This set of SDFs (or "level set") allowed the calculation of the distribution of each voxel's volume over the 5 compartments [46]. This layer volume distribution provided the basis for the laminar GLM discussed below.

## Extraction of layer-specific time courses

Since our fMRI data consisted of 0.8-mm isotropic voxels, they will almost certainly contain signals from several layers. Therefore, simply interpolating the fMRI signal at different depths will lead to contamination from neighbouring layers. To deal with this partial volume problem, we decomposed the layer signals by means of a spatial GLM [32,34,38,46]. For each ROI, a laminar $n \times k$ design matrix $\mathbf{X}$ represents the layer volume distribution, i.e., the distribution of the $k$ layers over the $n$ voxels within the ROI. Every row of $\mathbf{X}$ gives the distribution of a given

voxel volume over the layers, and every column (regressor) represents the volume of the corresponding layer across voxels. This laminar design matrix can be used in a spatial GLM to separate the voxels' BOLD signal for each of the 5 compartments through ordinary least squares (OLS) regression [46]:

$$\mathbf{Y} = \mathbf{X} \cdot \mathbf{B} + \boldsymbol{\varepsilon}$$

$\mathbf{Y}$ is a vector of voxel values from an ROI, $\mathbf{X}$ is the laminar design matrix, and $\mathbf{B}$ is a vector of layer signals. For each ROI and each functional volume, the layer signal $\hat{\mathbf{B}}$ was estimated by regressing $\mathbf{Y}$ against $\mathbf{X}$, yielding 5 depth-specific time courses per ROI and functional run.

In order to confirm that the method correctly identified GM, we quantified the raw signal in the EPI volumes for each of the 3 GM layers, as well as WM and CSF. As expected, the signal intensity was higher in the 3 GM layers (deep: 455 +/− 80; middle: 456 +/− 86; superficial: 461 +/− 80; mean +/− SD over participants) than in WM (427 +/− 83; lower than mean GM: $t_{17}$ = 5.17, $p = 7.7 \times 10^{-5}$) and outside of the brain (430 +/− 78; lower than mean GM: $t_{17}$ = 3.81, $p = 0.0014$).

In a control analysis, we extracted laminar time courses using an interpolation method rather than a spatial GLM. Interpolating a volume at different cortical depths effectively weights for all voxels in an ROI with respect to the layers:

$$\hat{\mathbf{B}}_{interpolation} = \mathbf{X}^{\mathrm{T}} \cdot \mathbf{Y}/N$$

$N$ is the number of voxels. The results of this analysis qualitatively replicated those of the main analysis and are presented in S3 Fig.

## Estimating effects of interest per layer

We estimated the effects of interest in the 3 GM layers using a temporal GLM (S6 Fig). Regressors of interest (presentation/omission of 45˚ or 135˚ gratings, separately for orientation and contrast task runs) were constructed by convolving stick functions representing the onsets of the trials with SPM12's canonical haemodynamic response function. Regressors of no interest included head motion parameters, their derivatives, and the square of the derivatives. Both the data and design matrix were high-pass filtered (cutoff = 128 seconds) to remove low-frequency signal drifts. The GLM explained close to half of the variance in each cortical layer (deep: $R^2$ = 0.43 +/− 0.09, middle: $R^2$ = 0.44 +/− 0.08, superficial: $R^2$ = 0.46 +/− 0.09; mean +/− SD over participants). The parameter estimates for the regressors of interest were the basis of our main analyses, described below.

To calculate orientation-specific BOLD responses for each ROI (e.g., the 45˚-preferring V1 ROI), the estimated BOLD response for conditions in which the non-preferred orientation was presented/expected (e.g., a 135˚ expected-but-omitted grating) was subtracted from the response for the corresponding condition in which the preferred orientation was presented/expected (e.g., a 45˚ expected-but-omitted grating). After this subtraction, responses were averaged over the 2 V1 ROIs, yielding layer-specific orientation-specific BOLD responses to each of the conditions of interest (expected-and-presented and expected-but-omitted gratings per task). These estimated BOLD responses were subjected to a 3-way repeated measures ANOVA with factors stimulus type (presented versus omitted), cortical layers (deep, middle, and superficial), and task (orientation versus contrast). Our main effect of interest, namely whether laminar BOLD profiles differed for presented and expected-but-omitted stimuli, was tested by the interaction of stimulus type (presented versus omitted) and cortical layer (deep, middle, and superficial). The 3-way interaction of stimulus type, layer, and task tested whether expectation effects were task dependent. To follow up a significant 3-way interaction, we

conducted 2-way repeated-measures ANOVAs with factors task (orientation versus contrast task) and cortical layer (deep, middle, and superficial) separately for the "stimulus present" and "stimulus omitted" conditions. Significant interactions were followed up with paired-sample *t* tests. To visualise the relevant across-subject variance for the within-subject ANOVA, errors bars in all figures show within-subject standard error of the mean (SEM) [89,90].

## Supporting information

**S1 Fig. Layer-specific BOLD response in V1 for presented and expected stimuli separately for voxel subpopulations preferring (solid lines, filled shapes) and non-preferring (dashed lines, open shapes) the current orientation.** BOLD responses are higher in subpopulations preferring the (expected) orientation in all layers for presented stimuli and deep layers only for expected-but-omitted stimuli. Note that the omission responses are overall negative. This is likely the result of the fact that the current study employed a fast event-related design without an explicit baseline period. Specifically, in this type of design, the baseline is effectively the mean signal, and when a stimulus is omitted, during a run in which stimuli are presented most of the time, the signal in V1 is likely to be lower than average. Essentially, this type of design is optimal for detecting differences between conditions (stimulus vs. omission or 45˚ stimulus/omission vs. 135˚ stimulus/omission), which was our main interest here, but suboptimal for detecting main effects of single conditions (e.g., stimulus vs. baseline or omission vs. baseline). Dots represent individual participants, and curved shapes indicate density. Error bars indicate within-subject SEM. Data are available at osf.io/k54p3. BOLD, blood oxygen level–dependent; SEM, standard error of the mean; V1, primary visual cortex.
(TIF)

**S2 Fig. Layer-specific BOLD response in V1 for presented and expected stimuli based on raw voxel time courses.** Normalising and weighting voxel time courses by orientation selectivity was omitted in this control analysis. **(A)** Orientation-specific BOLD response to presented (blue) and expected-but-omitted (orange) gratings in the different layers of V1, averaged over tasks. **(B)** Orientation-specific BOLD response to expected-but-omitted (orange, top panel) and presented (blue, bottom panel) gratings, separately for the orientation (solid lines, filled shapes) and contrast (dashed lines, open shapes) tasks. Dots represent individual participants, and curved shapes indicate density. Error bars indicate within-subject SEM. Data are available at osf.io/k54p3. BOLD, blood oxygen level–dependent; SEM, standard error of the mean; V1, primary visual cortex.
(TIF)

**S3 Fig. Layer-specific BOLD response in V1 for presented and expected stimuli determined by interpolation rather than spatial GLM. (A)** Orientation-specific BOLD response to presented (blue) and expected-but-omitted (orange) gratings in the different layers of V1, averaged over tasks. **(B)** Orientation-specific BOLD response to expected-but-omitted (orange, top panel) and presented (blue, bottom panel) gratings, separately for the orientation (solid lines, filled shapes) and contrast (dashed lines, open shapes) tasks. Dots represent individual participants, and curved shapes indicate density. Error bars indicate within-subject SEM. Data are available at osf.io/k54p3. BOLD, blood oxygen level–dependent; GLM, general linear model; SEM, standard error of the mean; V1, primary visual cortex.
(TIF)

**S4 Fig. Layer-specific BOLD response in V1 for presented and expected as a function of the number of selected voxels. (A)** Orientation-specific BOLD response to expected-but-omitted gratings in the different layers of V1, averaged over tasks. **(B)** Orientation-specific

BOLD response to presented gratings in the different layers of V1, averaged over tasks. Error bars indicate within-subject SEM. Data are available at osf.io/k54p3. BOLD, blood oxygen level–dependent; SEM, standard error of the mean; V1, primary visual cortex.
(TIF)

**S5 Fig. Registration of cortical boundaries to mean EPI for all participants.** Registrations are shown after rigid-body registration only (BBR), as well as after RBR. RBR increased absolute GM–WM contrast (c) in all participants. Arrows highlight locations where RBR improved registration. BBR, boundary-based registration; EPI, echo planar imaging; GM, grey matter; RBR, recursive boundary registration; WM, white matter.
(TIF)

**S6 Fig. Illustration of temporal GLM method.** Example model and data shown for 1 participant (P1) and 1 ROI (V1, 135 degree preferring voxels). Left, top panel: regressors used in the temporal GLM. Coloured time courses indicate regressors for the 4 conditions of interest, and grey time courses indicate nuisance regressors (i.e., head motion). Left, 3 bottom panels: fMRI time courses in each of the 3 GM layers (solid grey) and time courses fit by GLM (dashed black). Right, 3 bottom panels: parameter estimates for the 4 regressors of interest, quantifying the amplitude of the BOLD response evoked by the 4 conditions. These parameter estimates constitute the main results as shown in Fig 3. Data are available at osf.io/k54p3. BOLD, blood oxygen level–dependent; fMRI, functional magnetic resonance imaging; GLM, general linear model; GM, grey matter; ROI, region of interest.
(TIF)

## Acknowledgments

The authors would like to thank the Wellcome Centre for Integrative Neuroimaging radiography team for assistance with data collection and Nancy Rawlings for coordinating project administration. We are grateful to Stephen Fleming and Floris de Lange for comments on an earlier version of the manuscript.

## Author Contributions

**Conceptualization:** Peter Kok.

**Data curation:** Fraser Aitken, Georgios Menelaou, Renée S. Koolschijn.

**Formal analysis:** Fraser Aitken, Oliver Warrington, Peter Kok.

**Funding acquisition:** Peter Kok.

**Methodology:** Oliver Warrington, Nadège Corbin, Martina F. Callaghan.

**Project administration:** Fraser Aitken.

**Supervision:** Peter Kok.

**Writing – original draft:** Fraser Aitken, Peter Kok.

**Writing – review & editing:** Fraser Aitken, Georgios Menelaou, Oliver Warrington, Renée S. Koolschijn, Nadège Corbin, Martina F. Callaghan, Peter Kok.

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
