## [Editor Report · Decision Letter 0]

4 Sep 2020

Dear Dr Kok, 

Thank you for submitting your revised manuscript entitled "Prior expectations evoke stimulus-specific activity in the deep layers of V1" for consideration as a Research Article by PLOS Biology.

Your revisions have now been evaluated by the PLOS Biology editorial staff, as well as by the academic editor, and I'm writing to let you know that we would like to send your revised submission out for re-review.

However, before we can send your manuscript back out to reviewers, we need you to complete your submission by providing the metadata that is required for full assessment. To this end, please login to Editorial Manager where you will find the paper in the 'Submissions Needing Revisions' folder on your homepage. Please click 'Revise Submission' from the Action Links and complete all additional questions in the submission questionnaire.

Please re-submit your manuscript within two working days, i.e. by Sep 08 2020 11:59PM.

Kind regards,

Roli Roberts

Senior Editor

PLOS Biology

---

## [Decision Letter · Decision Letter 1]

16 Oct 2020

Dear Dr Kok,

Thank you very much for submitting a revised version of your manuscript "Prior expectations evoke stimulus-specific activity in the deep layers of V1" for consideration as a Short Report at PLOS Biology. This revised version of your manuscript has been evaluated by the PLOS Biology editors, the Academic Editor and the original reviewers.

In light of the reviews (below), we are pleased to offer you the opportunity to address the remaining points from the reviewers in a revised version that we anticipate should not take you very long. We will then assess your revised manuscript and your response to the reviewers' comments and we may consult the reviewers again.

We expect to receive your revised manuscript within 1 month.

**IMPORTANT - SUBMITTING YOUR REVISION**

*Resubmission Checklist*

*Published Peer Review*

*PLOS Data Policy*

*Blot and Gel Data Policy*

Sincerely,

Roli Roberts

Senior Editor,

rroberts@plos.org,

PLOS Biology

REVIEWERS' COMMENTS:

Reviewer #1:

[identifies herself as Cheryl A. Olman]

Thank you for adding the additional details about data handling -- the revised manuscript gives the reader a good sense of the data. 

The illustration of the anatomical landmarks and regions of interest (selected voxels) in Fig. 2 is very helpful. The legend should probably include a note that the red and yellow lines are the pial and wm surfaces of the GM after warping via RBR to the functional data. It's clear from this figure that RBR struggled with the contrast in the EPI and sometimes underestimated the cortical thickness, but I think this is a challenge confronted by the entire field, and the additional analyses provided in the main ms and the supplemental figures provide reasonable confidence that the unavoidable imperfections in registration/GM definition did not produce systematic bias.

The additional figure created to support the review process, which shows the negative BOLD response to the omitted stimuli, helps a great deal with interpretation and understanding how analysis was done. I think including it in the supplemental material, along with the excellent explanation provided for the review, is a good idea. 

Reviewer #2:

In my review of an earlier version of the manuscript, I had raised concerns that the presentation was too abbreviated to judge the merits of the experiment based on the data. My comments were echoed by two other reviewers. For this revision of the manuscript, the authors went through substantial amount of effort to address my main concern (that presentation was too condensed for reviewers / readers). I think the work that went into the revision has made it a much better paper.

R1 and R3 raised additional, more detailed methodological questions, which are carefully addressed in this revision. As a result, the manuscript now lays out a much clearer and easier to understand logic. Crucially, it includes much more detail about the dataset and intermediate processing steps, which previously had to be inferred from related publications. 

At this point, I have no further concerns.

Reviewer #3:

The authors have responded to my initial comments satisfactorily. However, I do have a concern about one issue raised by Reviewer 1, point 4 regarding the voxel selection (and I apologize for not raising this point in the initial review). The authors new analysis rule out a bias due to unequal sampling of different laminae, but there is another potential issue which is to do with the spatial distribution of voxels on the cortical surface. Assuming Fig 2A is representative, it is clear that only a tiny fraction of activated voxels are included among the selected voxels, and these appear to be scattered widely across the V1 ROI. Given that the conclusions hinge on the ability to assign differences in responses solely to laminar position, the authors need to demonstrate that the voxels from each layer are distributed randomly across the surface and not clustered in some systematic way (e.g. by eccentricity or angular position). This could be done for instance by plotting the voxels on the cortical surface and labeling them by laminar position and check this plot for clustering (supported by a suitable statistical procedure). Alternatively, they could demonstrate that any variations in response strength along the cortical surface across all layers could not account for the effects seen. Or, better still, they could redo the laminar analysis including increasingly large numbers of voxels and show that the results remained qualitatively similar. There is considerable evidence that orientation preference in V1 is driven by large-scale spatial patterns rather than columnar sampling (this is actually evident in Fig 2A, where voxels tuned to the different orientations are spatially clustered) so lowering the threshold for voxel inclusion should not be expected to have dramatic effects on orientation tuning of the voxels. As Reviewer 1 points out, the voxel selection method is very restrictive and it needs to be shown that the results are not influenced by this procedure. 

Reviewer #4:

The authors have addressed my comments related to data analyses and statistics and I believe the results presented in the revised manuscript are stronger.

The authors explain the difference between expectation and attention/working memory. It would be helpful if the authors discussed the role of mental imagery and how it relates to expectation. Recent work suggests that mental imagery relates to BOLD in deeper layers, similar to the results for expectation presented by the authors.

---

## [Editor Report · Decision Letter 2]

11 Nov 2020

Dear Dr Kok,

Thank you for submitting your revised Short Report entitled "Prior expectations evoke stimulus-specific activity in the deep layers of V1" for publication in PLOS Biology. I have now assessed your revisions and discussed them with the Academic Editor. 

Based on this assessment, we're delighted to let you know that we're now editorially satisfied with your manuscript. However before we can formally accept your paper and consider it "in press", we also need to ensure that your article conforms to our guidelines. A member of our team will be in touch shortly with a set of requests. As we can't proceed until these requirements are met, your swift response will help prevent delays to publication. Please also make sure to address the data and other policy-related requests noted at the end of this email.

IMPORTANT:

a) Please could you change your title to "Prior expectations evoke stimulus-specific activity in the deep layers of the primary visual cortex"?

b) Please attend to my Data Policy and Ethics Policy requests at the foot of the email.

- a cover letter that should detail your responses to any editorial requests, if applicable

*Copyediting*

*Published Peer Review History*

*Early Version*

Sincerely,

Roli Roberts

Senior Editor,

rroberts@plos.org,

PLOS Biology

ETHICS STATEMENT:

-- Please include information about the form of consent (written/oral) given for research involving human participants. All research involving human participants must have been approved by the authors' Institutional Review Board (IRB) or an equivalent committee, and all clinical investigation must have been conducted according to the principles expressed in the Declaration of Helsinki.

DATA POLICY:

Regardless of the method selected, please ensure that you provide the individual numerical values that underlie the summary data displayed in the following figure panels as they are essential for readers to assess your analysis and to reproduce it: Figs 2A, 3AB, S1, S2AB, S3AB, S4AB, S5, S6. NOTE: the numerical data provided should include all replicates AND the way in which the plotted mean and errors were derived (it should not present only the mean/average values).

---

## [Editor Report · Decision Letter 3]

20 Nov 2020

Dear Dr Kok,

On behalf of my colleagues and the Academic Editor, Adam Kohn, I am pleased to inform you that we will be delighted to publish your Short Reports in PLOS Biology. 

PRODUCTION PROCESS

Before publication you will see the copyedited word document (within 5 business days) and a PDF proof shortly after that. The copyeditor will be in touch shortly before sending you the copyedited Word document. We will make some revisions at copyediting stage to conform to our general style, and for clarification. When you receive this version you should check and revise it very carefully, including figures, tables, references, and supporting information, because corrections at the next stage (proofs) will be strictly limited to (1) errors in author names or affiliations, (2) errors of scientific fact that would cause misunderstandings to readers, and (3) printer's (introduced) errors. Please return the copyedited file within 2 business days in order to ensure timely delivery of the PDF proof. 

If you are likely to be away when either this document or the proof is sent, please ensure we have contact information of a second person, as we will need you to respond quickly at each point. Given the disruptions resulting from the ongoing COVID-19 pandemic, there may be delays in the production process. We apologise in advance for any inconvenience caused and will do our best to minimize impact as far as possible.

EARLY VERSION

PRESS 

Kind regards,

Alice Musson

Publishing Editor, 

PLOS Biology

on behalf of

Roland Roberts,

Senior Editor

PLOS Biology